# Structure of the EphB6 receptor ectodomain

**Emilia O. Mason**[1], **Yehuda Goldgur**[1], **Dorothea Robev**[1], **Andrew Freywald**[2], **Dimitar B. Nikolov**[1]*, **Juha P. Himanen**[1]*

1 Structural Biology Program, Memorial Sloan Kettering Cancer Center, New York, New York, United States of America, 2 Department of Pathology and Laboratory Medicine, University of Saskatchewan, Saskatoon, Canada

* nikolovd@mskcc.org (DBN); himanenj@mskcc.org (JPH)

**Data Availability Statement:** All relevant data are within the paper and its Supporting Information files.

**Funding:** D.B.N., R01-NS038486, National Institutes of Health, www.nih.gov; APS, P41 GM103403 and DE-AC02- 06CH11357, NIH and U.

## Abstract

Eph receptors are the largest group amongst the receptor tyrosine kinases and are divided into two subgroups, A and B, based on ligand binding specificities and sequence conservation. Through ligand-induced and ligand-independent activities, Ephs play central roles in diverse biological processes, including embryo development, regulation of neuronal signaling, immune responses, vasculogenesis, as well as tumor initiation, progression, and metastasis. The Eph extracellular regions (ECDs) are constituted of multiple domains, and previous structural studies of the A class receptors revealed how they interact with ephrin ligands and simultaneously mediate Eph-Eph clustering necessary for biological activity. Specifically, EphA structures highlighted a model, where clustering of ligand-bound receptors relies on two distinct receptor/receptor interfaces. Interestingly, most unliganded A class receptors also form an additional, third interface, between the ligand binding domain (LBD) and the fibronectin III domain (FN3) of neighboring molecules. Structures of B-class Eph ECDs, on the other hand, have never been reported. To further our understanding of Eph receptor function, we crystallized the EphB6-ECD and determined its three-dimensional structure using X-ray crystallography. EphB6 has important functions in both normal physiology and human malignancies and is especially interesting because this atypical receptor innately lacks kinase activity and our understanding of the mechanism of action is still incomplete. Our structural data reveals the overall EphB6-ECD architecture and shows EphB6-LBD/FN3 interactions similar to those observed for the unliganded A class receptors, suggesting that these unusual interactions are of general importance to the Eph group. We also observe unique structural features, which likely reflect the atypical signaling properties of EphB6, namely the need of co-receptor(s) for this kinase-inactive Eph. These findings provide new valuable information on the structural organization and mechanism of action of the B-class Ephs, and specifically EphB6, which in the future will assist in identifying clinically relevant targets for cancer therapy.

## 1. Introduction

Human cells express 14 receptor tyrosine kinases (RTKs) of the Eph type, which constitute the largest group within the RTK family [1]. The structural organization of Eph receptors is typical for most RTKs, they are single peptide chain proteins with a singular transmembrane spanning

S. DOE, www.grants.gov. The funders had no role
in study design, data collection and analysis,
decision to publish, or preparation of the
manuscript.

**Competing interests:** The authors have declared
that no competing interests exist.

helix. The extracellular region of Eph receptors includes an N-terminal ligand-binding domain, followed by a cysteine-rich region and two fibronectin-type-3 domains. The intracellular region contains a juxtamembrane region, which has a regulatory function, a tyrosine kinase domain, a SAM domain and a PDZ-binding motif [2, 3] (Fig 1). Eph receptors are separated into EphA and EphB classes based on their sequence similarity and ligand-binding preferences, where EphA receptors (EphA1 –EphA8 and EphA10) predominantly interact with ephrin-A ligands (ephrin-A1 –ephrin-A6) that are GPI-anchored to the cell membrane, while EphB receptors (EphB1 –EphB4 and EphB6) preferentially bind transmembrane ligands of the ephrin-B type (ephrin-B1 –ephrin-B3) [4]. The biological activities of Eph receptors are very diverse and through their signaling they control many aspects of cell behavior, including cell-cell and cell-matrix interactions, cell motility, as well as cell survival and proliferation [5, 6]. Consistent with this, Eph receptors are actively involved in governing a large variety of responses during embryo development, in adult physiology and in pathological conditions [1, 6, 7]. Several members of the Eph receptor family are overexpressed in tumors [8] and promote tumor aggressiveness [9]. Interestingly high expression levels of Ephs in cancer cells often correlate with low levels of kinase-domain phosphorylation, suggesting that their oncogenic activities could be mediated by nonconventional signaling mechanisms [6]. The signaling activity of Eph receptors is determined by the formation of pre-existing and ligand-induced clusters [3, 10]. While ligand-triggered oligomerization is initiated by the Eph ligand-binding domains [10], the overall signaling is also modulated by the interactions between several other structural motifs within the cysteine-rich region, the Eph fibronectin-type-3 domains, and the transmembrane domain [10–16]. Overall, the organization of Eph receptors in various types of cell-surface clusters or oligomers is responsible for both activation of their catalytic function and for defining the precise signaling outputs.

Generally Eph receptor signaling relies on their kinase activity [4, 17], but both EphA and EphB classes contain innately kinase-deficient members, EpA10 and EphB6, that lack catalytic activity because of alterations of conserved motifs within their kinase domains [18]. The presence of kinase-inactive receptors in each Eph class suggest that these molecules are important participants in the Eph receptor signaling network. While relatively little is known about the EphA10 biology, the EphB6 receptor is known to have important functions in both normal physiology and malignancy. For example, EphB6 has been reported to modulate T cell responses [19–22] and to control blood pressure in male mice [23, 24]. The role of EphB6 in cancer is quite complex, as it not only downregulates and suppresses aggressiveness in multiple malignancies and cancer cells [25–47], but also supports tumor initiation in colorectal and breast cancers [45, 48]. Due to the absence of the intrinsic kinase activity, EphB6 often operates by interacting with kinase-active Eph receptors and these interactions likely determine the signaling outcomes in varying biological contexts [32, 49, 50]. Although structures of kinase-active Eph receptors have been extensively characterized [2, 4, 10, 17, 51, 52], structural characterization of kinase-dead Eph receptors has never been reported, which significantly limits our ability to understand the mechanisms of their signaling or their interactions with other Eph receptors. Moreover, all reported Eph ectodomain structures have been of the A-class receptors. Therefore, we determined and analyzed in detail the structural organization of the extracellular domain of the EphB6 receptor (EphB6-ECD).

## 2. Methods and materials

### 2.1 Materials

EphB6 custom oligo primers were purchased from Fisher Scientific Life Sciences. The Anza T4 DNA Ligase Master Mix kit was purchased from Invitrogen. The NucleoSpin Gel and PCR

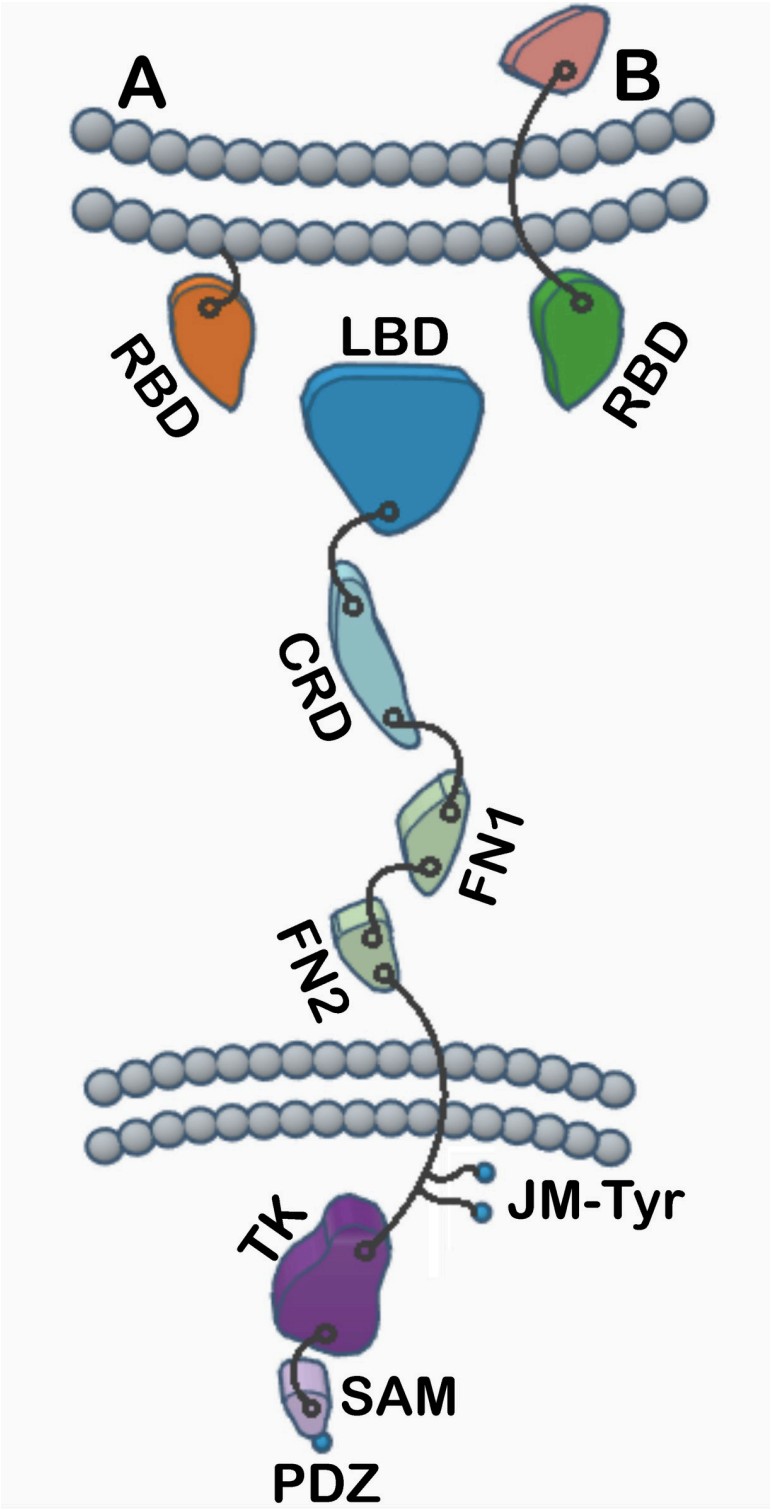

**Fig 1. Domain organization of Eph receptors and ephrin ligands.** Class A and class B receptor-binding domains (RBDs), ligand-binding domain (LBD), cysteine-rich domain (CRD), fibronectin-type-3 region (FN1 and FN2), juxtamembrane tyrosine residues (JM-Tyr), tyrosine kinase (TK), sterile alpha motif (SAM), PDZ-binding motif (PDZ).

clean-up kit was purchased from Takara Bio USA, Inc. The Autorisierter Thermal Cycler PCR instrument was purchased from Eppendorf. Lipofectamine 2000 transfection reagent and Hygromycin B were purchased from Sigma Aldrich. Nalgene filtration systems along with the restriction enzymes Nhe1 and BamH1 were purchased from Thermo Scientific. Protein A Sepharose Fast Flow beads were purchased from GE Healthcare Bio-Sciences AB and the columns used for packing the beads were from Pharmacia Biotech. Both the PowerPAC 300 2-D gel instrumentation and the 4–20% 20 μL well Mini-PROTEAN TGX gels were purchased from Bio-RAD. Instant Blue, used to stain the gels, was purchased from Expedeon Ltd. The HEK 293 cell line was purchased from ATCC and the media DMEM, phosphate buffered saline, penicillin streptomycin, and fetal bovine serum were purchased from the Memorial Sloan Kettering Cancer Center (MSKCC) Media Preparation Facility Center. One liter roller bottles and trypsin EDTA (1X) were purchased from Fischer Scientific. Cell culture dishes were purchased from Greiner. Agarose gels were purchased from FMC Bioproducts, thrombin was purchased from Novagen, and 10 K MWCO Spin-X UF concentrators were purchased from Corning. The Fast Protein Liquid Chromatography (FPLC) SD 200 size-exclusion column (Amersham Biosciences Corp.) was used for purifying samples, and a TTP Labtech Mosquito instrument was used for crystal plating. Salt Rx 1 and 2 crystal screens along with MCR 2 Well Crystallization Plates were purchased from Hampton Research.

## 2.2 EphB6 constructs

Initially, three cDNA fragments were produced by PCR, encoding various fragments of the extracellular region of human EphB6: 18–567 (DNA sequence 3' to 5': `TTT ATT GGA TCC GAT CAC CAA GGA GAG TC`), 18–573 (DNA sequence 3' to 5': `TAT TAT GGA TCC GGA GCC GAT CAC CAA GG`), and 18–579 (DNA sequence 3' to 5': `TTT ATT GGA TCC AGC CCC CAG GAT GGA GCC G`). These fragments were cloned into a lab made version of the pcDNA 3.1 hygromycin vector encoding also the Fc portion of the human IgG [53] to generate EphB6-ECD-Fc fusions. Between the EphB6 insert and the Fc region, these fusions also carry a thrombin cleavage site that allows the removal of the Fc tag. Briefly, Nhe1 and BamH1 restriction enzymes were used to digest the vector and the cDNA fragments. The digested products were purified using a 1% agarose gel followed by gel extraction utilizing both the NucleoSpin Gel and PCR clean-up kit. The purified cDNA fragments were ligated into the vector using the Anza T4 DNA Ligase Master Mix kit. Of the three constructs, both 1–584 and 1–589 expressed well, with 1–584 having a higher level of expression and it was, therefore, selected for further experiments.

## 2.3 Expression in HEK 293 cells and purification

HEK293 cells were grown in DMEM and supplemented with 10% (vol/vol) FBS, 1000 units/ mL penicillin, and 100 μg/mL streptomycin. Cells were transfected with the EphB6 construct at 80–90% confluence in six-well plates using Lipofectamine 2000, and hygromycin was used for selecting resistant cell populations. The obtained HEK293 colonies were assessed for EphB6 expression levels and the high expression colonies were further expanded, as well as frozen for preservation. Large scale EphB6-ECD-Fc production was performed using 1 L roller bottles. These bottles allow high-density growth at 37˚C. After 3 days, the EphB6-ECD-Fc containing media was harvested and filtered, and the secreted fusion protein was purified using a Protein A Sepharose column. The protein was eluted from the beads with 100 mM glycine at pH 3. The Fc tag was removed using thrombin cleavage overnight at +4° C. Further Fc cleanup was performed with Protein A Sepharose beads and the EphB6-ECD in the supernatant was concentrated using a 10K MWCO concentrator. The concentrated protein was next purified

on a FPLC SD200 size-exclusion column. These purification steps were monitored by 2D-gel electrophoresis using 4–20%, 20 μL/well gels. The extracellular region (residues S-28 to G-181) of murine ephrin-B2 was expressed as an Fc-fusion protein in HEK293 cells from a modified pcDNA3.1 vector (Invitrogen) containing a CD4 signal sequence as described in [54]. Binding kinetics were measured by Biolayer Interferometry (BLI) on a BLItz instrument (ForteBio) as described in [55]. Specifically, the Protein-A biosensor was loaded with 50 μg/ml solution of purified Fc-tagged ephrin-B2, the sensor was washed with HBS and the association and dissociation measurements for the EphB6-ECD at 100 nM and 1 μM were carried out for 3 min each. Kinetic parameters ($k_{on}$ and $k_{off}$) and affinities ($K_d$) were calculated using the BLItzPro software.

## 2.4 Crystal screens of EphB6-ECD and data collection

The purified EphB6-ECD protein was concentrated to 10 mg/mL in Hepes-buffered saline solution and crystal trays were set with a Mosquito instrument using various well solution conditions at 20˚C. EphB6-ECD was crystallized using sitting drop vapor diffusion against a well solution containing 1.5 M sodium nitrate, 0.1 M sodium acetate trihydrate pH 4.6, which was from the Salt Rx 1 and 2 screens (Hampton Research). The crystals were frozen in a cryobuffer containing additional 25% (vol/vol) glycerol and analyzed at the Advanced Photon Source at Argonne National Laboratory in Chicago, IL. Data was processed using the HKL2000 package. The structure was by automatic molecular replacement using BALBES [56]. Refinement was performed with Phenix and interactive model building—with O.

## 3. Results and discussion

### 3.1 Purification and characterization of the EphB6-ECD

To analyze the architecture of the extracellular domain of the human EphB6 receptor, we determined its crystal structure. EphB6-ECD was expressed as an Fc fusion protein in HEK-293 cells and purified by Protein-A Sepharose affinity and Superdex-200 size-exclusion chromatography (Fig 2) to obtain over 90% pure preparation with the protein of a molecular weight around 65 kDa (Fig 2C). This preparation was used for crystallization (Fig 2D).

EphB6-ECD binding studies were performed with an ephrin-B class ligand to determine, whether the protein was biologically active. Specifically, the ephrin-B2-Fc ligand was loaded onto the Protein A tip in the Blitz instrument and its interaction with EphB6-ECD was evaluated. As shown in Fig 3A, EphB6 bound ephrin-B2 with an apparent $K_d$ of 89 nM, a somewhat lower affinity than that for ephrin-B2 binding to EphB2 (8.6 nM) [57]. The Superdex size-exclusion chromatography profile shows that the EphB6-ECD/ephrin-B2 complex has a 1:1 stoichiometry at the micromolar concentration range used (Fig 3B).

### 3.2 Overall structure of EphB6-ECD

X-ray diffraction data from the EphB6-ECD crystals were collected at the Advanced Photon Source ID24-C beamline (Argonne National Laboratory, Chicago, IL) and the structure (PDB ID 7K7J) was determined using molecular replacement and refined to 3 Å resolution. Data collection and refinement statistics are summarized in Table 1.

The final refined structure of the EphB6 ECD is shown in Fig 4. This is the first reported structure of a B class Eph receptor ectodomain. Similar to EphA class ectodomain structures, it shows a rigid, extended, rod-like architecture with limited flexibility between the three domains: the ligand-binding domain (LBD), the cysteine-rich domain (CRD), and the Fibronectin III (FN3) domain (see Fig 1 for a schematic representation). N-terminal LBD has a

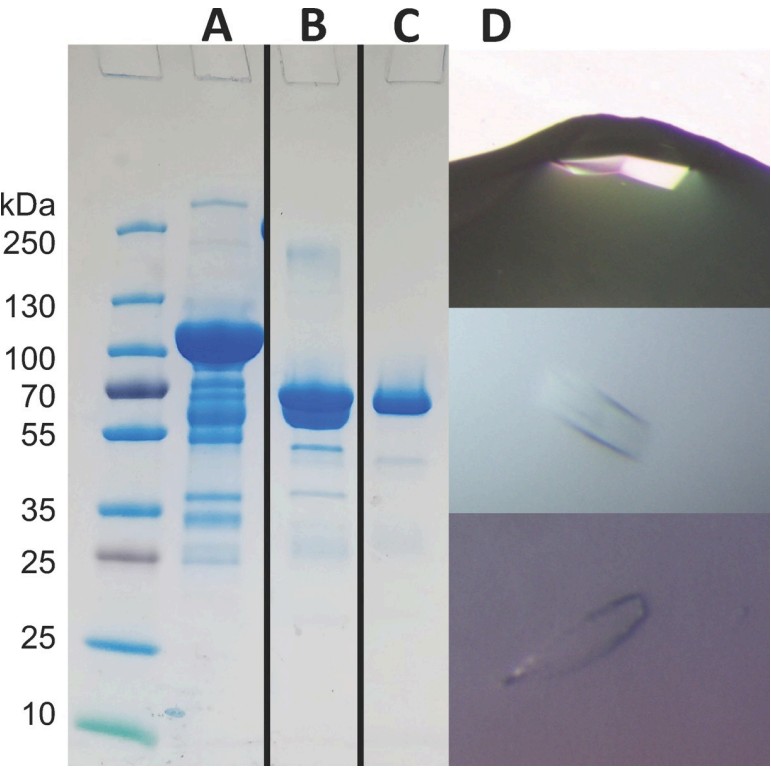

**Fig 2. Purification and crystallization of EphB6-ECD.** (A) EphB6-ECD-Fc after initial Protein A Sepharose purification; (B) EphB6-ECD upon addition of thrombin and Protein A Sepharose beads; (C) The finally purified EphB6-ECD, showing the expected molecular weight of ~65 kDa; (D) EphB6-ECD protein crystals.

jelly-roll folding topology with several loops of varying length packing against the two anti-parallel beta-sheets. The N-terminal part of the cysteine-rich domain includes antiparallel β-strands arranged as a β-sandwich, with several disulfide bonds stabilizing this region of the structure. As in the EphA2-ECD structure [11], the N- and C-terminal residues are located on opposite sides of the domain and the two CRD halves are packed against each other fairly tightly. The C-terminal half of the CRD contains β-strands and several closely packed random coils connected by disulfide bridges. Except for some negatively charged regions, the molecular surface is predominantly neutral. These structural features are, for the most part, very similar to those reported for the A-class receptors [11, 58]. The N-terminal FN3 domain adopts a typical immunoglobulin-like fold. The C-terminal FN3 domain is not clearly visible in the electron density map, similar to the EphA2-ECD crystal structure [11], consistent with its increased flexibility in relation to the rest of the Eph ectodomain.

### 3.3 Differences between the EphB6 and the EphA-class ECD structures

Although the overall structure of the EphB6-ECD is similar to those of the EphA2- and EphA4-ECDs, there are several structural differences with potential biological implications. The entire EphB6-ECD can be superimposed on the EphA2-ECD [11] with an rmsd value of 3.803 Å between the C-alpha atoms (Fig 5). EphB6-ECD can also be superimposed on the EphA4-ECD structure [58] with an rmsd value of 4.412 Å between the C-alpha atoms. These values are clearly higher than the rmsd value for the superimposition of EphA2 on EphA4 ECDs (2.052 Å). Further structural studies of other EphB receptors are needed to show if, indeed, all B class receptors are structurally more similar to each other than to the A class

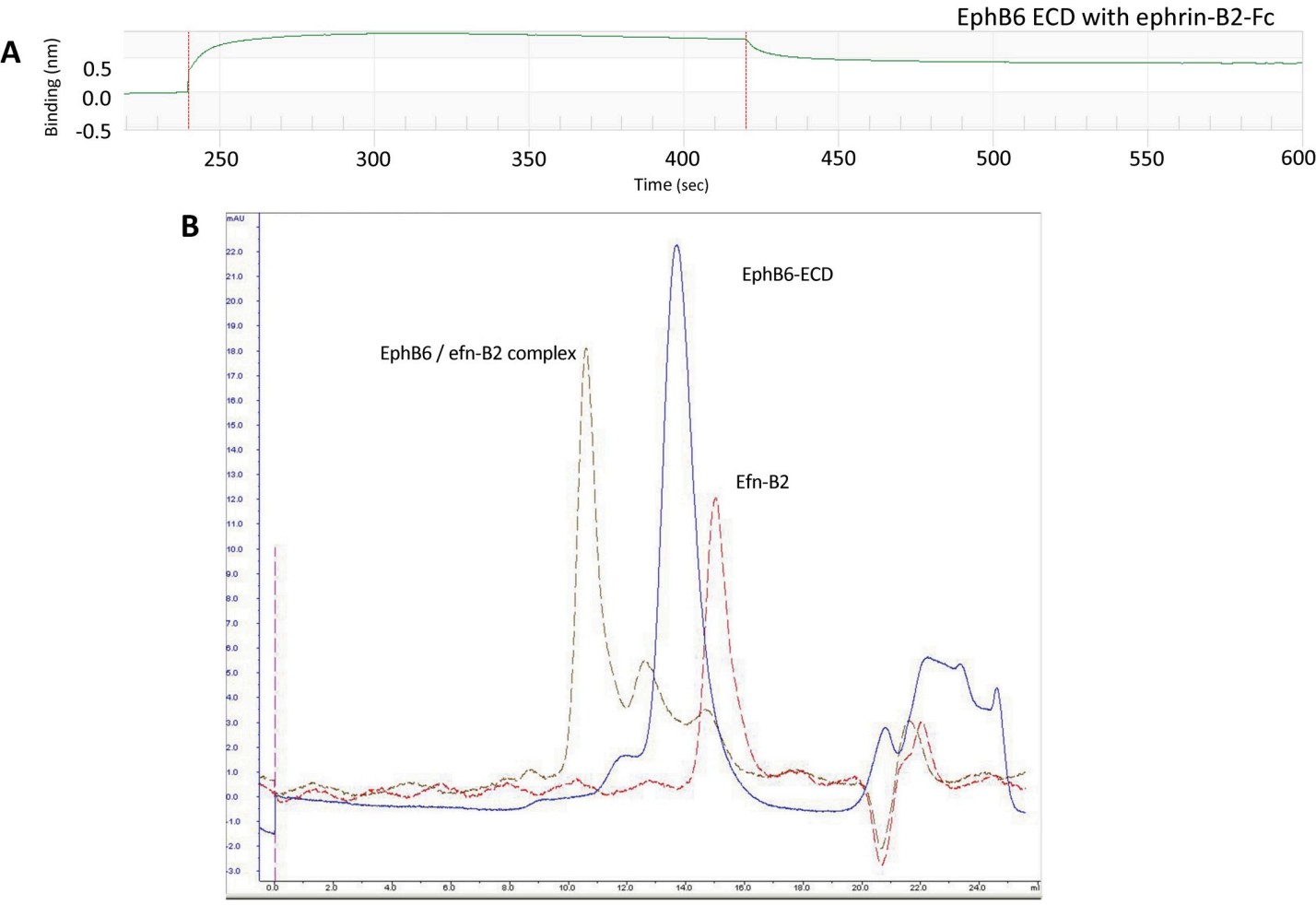

**Fig 3. Binding of the EphB6-ECD to the ephrin-B2-Fc ligand.** (A) EphB6-ECD binds to ephrin-B2-Fc with a $K_d$ value of 8.85 E$^{-8}$ M as measured on a Blitz instrument; (B) Size-exclusion FPLC on a Superdex200 column reveals a 1:1 stoichiometry for the EphB6-ECD/ephrin-B2 complex.

receptors and how this might reflect the somewhat different ligand binding and recognition properties of the Eph A and B class receptors [59–63].

When superimposed individually, the EphB6 and EphA2 domains have rmsd values between their C-alpha atoms of 0.592 Å for LBD, 0.870 Å for CRD, and 0.807 Å for FN. Several loops in the original EphA2 ECD structure [11] were not structured and consequently, could not be compared to their EphB6 counterparts. The lower rmsd values for the individual domains are indicative of the very high sequence and structural similarity of these conserved regions between the Eph receptors of the different classes, while the higher rmsd value for the entire ECDs implies flexibility of the linkers between the individual domains, despite the apparently rigid overall structural architecture.

When analyzed in detail there are several potentially functionally relevant structural differences between the EphB6 and EphA2 ECDs. First, inside the LBD, EphB6 has an insert of 11 consecutive Serine residues within the J-K loop (residues 151–161 in our construct). This "Super-Serine Loop" (our nomenclature) is not present in any other Eph molecule. Because the J-K loop is part of the ligand-binding module on the surface of the Eph receptors [64], it's plausible that the 11-Serine insert is partially responsible for the somewhat lower affinity of EphB6 for B ephrins, such as ephrin-B2, as compared, for example, to EphB2 (see above).

**Table 1. Crystallographic data and refinement statistics.**

| EphB6 ectodomain | |
|---|---|
| **Data collection** | |
| Beamline | APS 24-ID-C |
| Space group | $P4_32_12$ |
| Cell dimensions | |
| a, b, c (Å) | 105.3, 105.3, 188.7 |
| α, β, γ (°) | 90, 90, 90 |
| Resolution (Å) | 50–3.0 (3.05–3.0) |
| Wavelength (Å) | 0.9792 |
| $R_{pim}$ | 0.056 (0.546) |
| CC1/2 | 0.999 (0.603) |
| $<I>/<\sigma I>$ | 20.1 (1.6) |
| Completeness (%) | 99.9 (100.0) |
| Redundancy | 6.3 (6.7) |
| Unique reflections | 21919 |
| **Refinement** | |
| $R_{work}/R_{free}$ | 0.235/0.272 |
| *B*-factors (Å$^2$) Average/Wilson | 90.4/53.0 |
| RMS deviations | |
| bond lengths (Å) | 0.011 |
| bond angles (°) | 1.432 |
| Ramachandran plot | |
| % favored | 82.7 |
| % allowed | 16.4 |
| outliers | 0.9 |
| **Model contents** | |
| Protomers / ASU | 1 |
| Protein residues | 429 |
| Glycans | 2 |
| **PDB ID** | 7K7J |

Values in parentheses refer to the highest resolution shell. The $R_{free}$ set consists of 5% of the reflections chosen randomly against which the structure was not refined.

Indeed, this highly flexible serine insert might interfere with ligand binding by hindering the formation of a well-defined ligand-binding cavity on the receptor surface [60].

Second, the H-I loop, N-terminal to the J-K loop (residues 125–130 in our structure) represents another unique to EphB6 Serine-rich motif, which is fully structured (Fig 5B). This motif is not present in the EphA2-ECD and, indeed, this region is structurally considerably different between the two receptors: in EphB6 it forms an additional short loop, while in EphA2 the region is polar and flat. The S125-X-X-S128-X-S130 loop of EphB6 is not expected to participate in the initial ligand recognition and binding because it is located on the opposite side of the LBD, on the 'back' of the ligand-binding module. However, this surface area has been shown to participate in the formation of Eph/ephrin heterotetramers [51, 64]. Consequently, the presence of an additional loop (Fig 5B) in this 'heterotetramerization' [58] Eph-Eph interface would likely have an influence on the Eph-Eph interactions that mediate the formation of functional Eph/ephrin assemblies at the sites of cell-cell contact [51]. This might, in turn, affect the Eph-Eph cluster formation and dynamics and thus, the signaling characteristics.

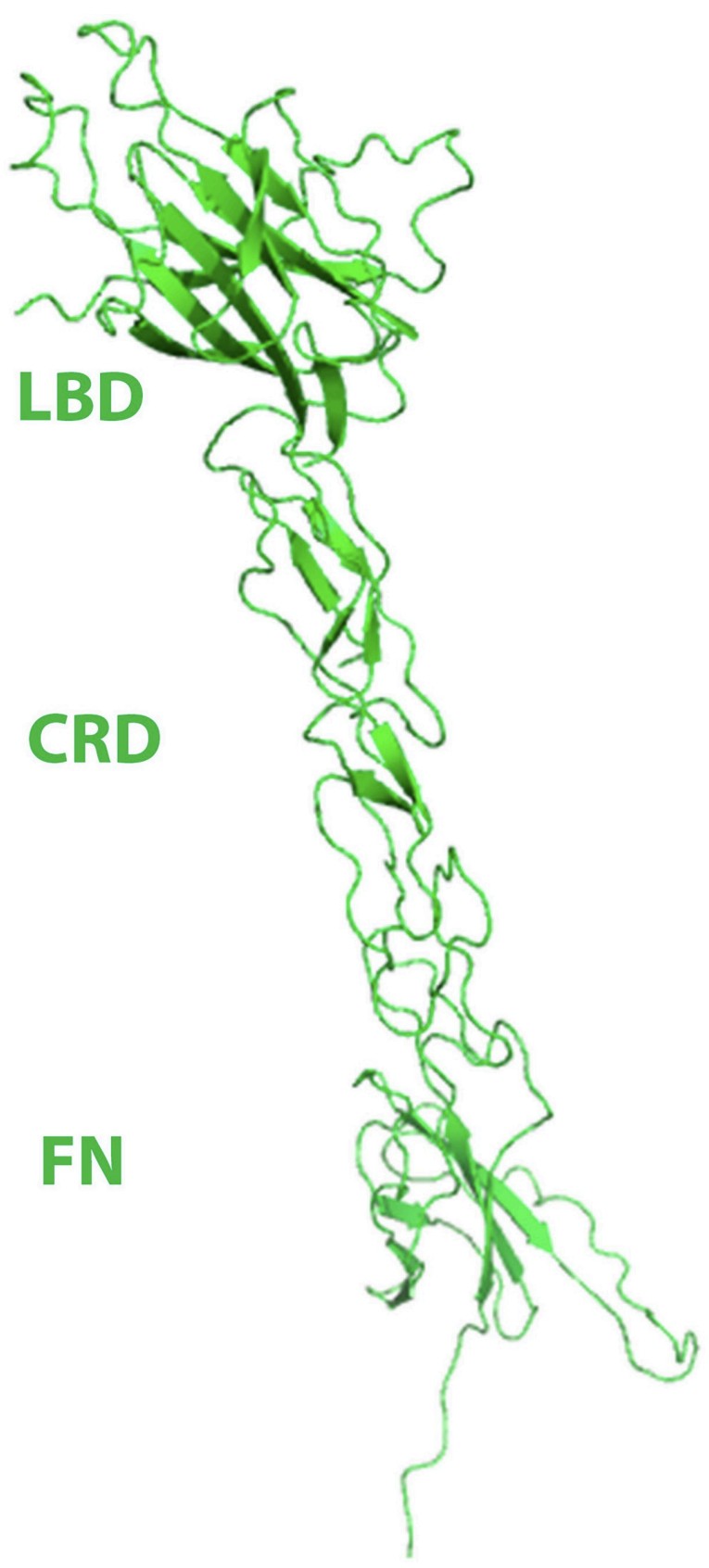

**Fig 4. EphB6-ECD structure showing the ligand binding (LBD), cysteine rich (CRD), and fibronectin (FN) domains.** The ectodomain has a rigid, rod-like conformation with limited flexibility between the subdomains. The LBD has a jelly-roll folding topology, while the CRD includes β-strands arranged as a β-sandwich, with several disulfide bonds stabilizing the structure. The N-terminal FN adopts a typical immunoglobulin-like fold.

Finally, the first FN domain of EphB6 has a seven-residue insertion in the loop between E-396 and G-406 and hence, adopts a significantly different conformation than the one observed in EphA2 (Fig 5C) or EphA4. There are no biological studies of the importance of this FN loop on Eph signaling. We suggest, though, that because EphB6 is kinase inactive and obligated to use other Ephs as co-receptors to effect signaling, this loop might be part of an Eph-Eph oligo-merization interface, to mediate interactions between EphB6 and other receptors (e.g. EphA2 [11]) within the same Eph/ephrin signaling assemblies. Indeed, the E396-G406 loop is in rela-tively close proximity to the cysteine-rich Eph domain (CRD), and to the so called "Eph-Eph clustering interface" [58, 65] and thus, might participate in the co-clustering of EphB6 recep-tors with other members of the Eph family. However, further structural studies are needed to substantiate this postulation.

### 3.4 Head-to-tail EphB6-ECD interactions

Earlier structural, cell biological and biochemical studies have shed light on the Eph signaling initiation mechanism [1]. Hence, a 'dimerization' interface within the LBD was shown to mediate the formation of receptor/ligand 1:1 hetero-dimers upon cell-cell contact, while two

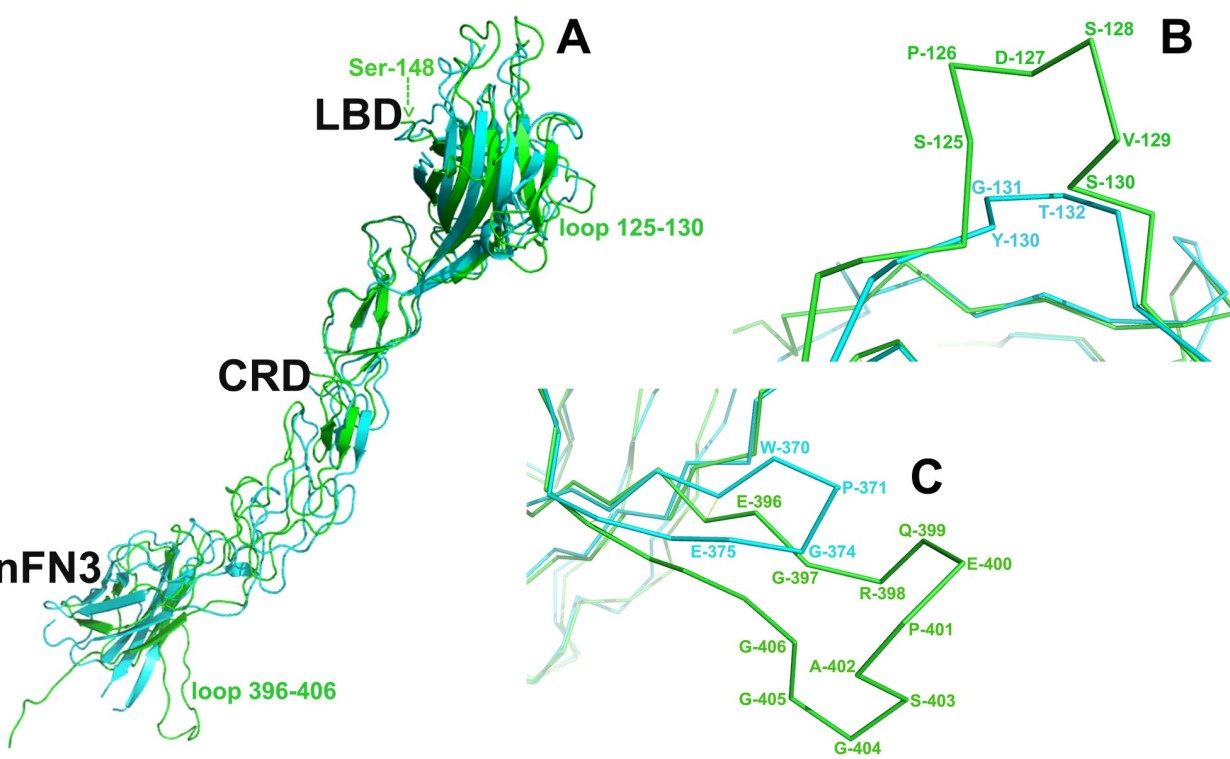

**Fig 5.** (A) Superimposition of the EphB6 and EphA2 ectodomains. The rmsd value between the C-alpha atoms is 3.803Å. The EphB6-ECD is colored in green and the EphA2-ECD—in cyan. Arrow shows the location of Ser-148 preceding the 'Super Serine' loop (residues 151–161) that is not visible in our electron density map. LBD, ligand-binding domain; CRD, cysteine-rich domain; nFN3, N-terminal fibronectin 3 domain. (B) Close-up of the superimposed LBD H-I loop of the two molecules (EphB6 residues 125–130). (C) Close-up of the superimposed FN domain loop (EphB6 residues 396–406) of the two molecules.

distinct Eph-Eph interfaces named "heterotetramerization" and 'clustering' mediate the formation of higher-order assemblies (clusters), necessary for full biological activation [4, 10]. However, recent structural studies have revealed that, in absence of ligands, another 'head-to-tail' Eph interface can be formed between the LBD of one unliganded Eph and the FN domain of a neighboring unliganded Eph molecule [11, 58]. The precise biological role of this 'pre-clustering' interface is currently being investigated and it has been proposed to be involved in the fine-tuning of Eph signaling. For example, the ligand-induced EphA4 signaling is decreased when the pre-clustering interface is weakened but increased if the interface is strengthened [58].

Our structural analysis shows that, similar to the A-class Eph-ECD structures published earlier [11, 58], the ectodomain of the unliganded EphB6 receptor also participates in LBD-FN3 head-to-tail interactions with its neighboring molecules (Fig 6A). The total buried area of this interface is 860 Å, close to that for EphA2 (980 Å) (unpublished data) but much smaller than the one reported for EphA4 (2,460 Å) (Fig 6B). It is quite possible that the considerably larger buried area in the pre-clustering interface of EphA4 might facilitate its promiscuity, allowing for an efficient activation of the highly pre-clustered receptors upon contact with any, either A- or B-class, ephrin ligand [66].

The EphB6 head-to-tail interface contains several well-defined van der Waals contacts and bonds, hydrogen bonds. The most prominent of the participating residues are Thr-35, Gly-39, and Ser-128 on the Eph LBD domain and Arg-413, Pro-458, and Pro-459 on the Eph FN domain (Fig 6B). On the other hand, the LBD/FN interface of EphA4, the only head-to-tail Eph interaction published so far, comprises of residues Cys-73, Asn-74, Val-75, Met-76,

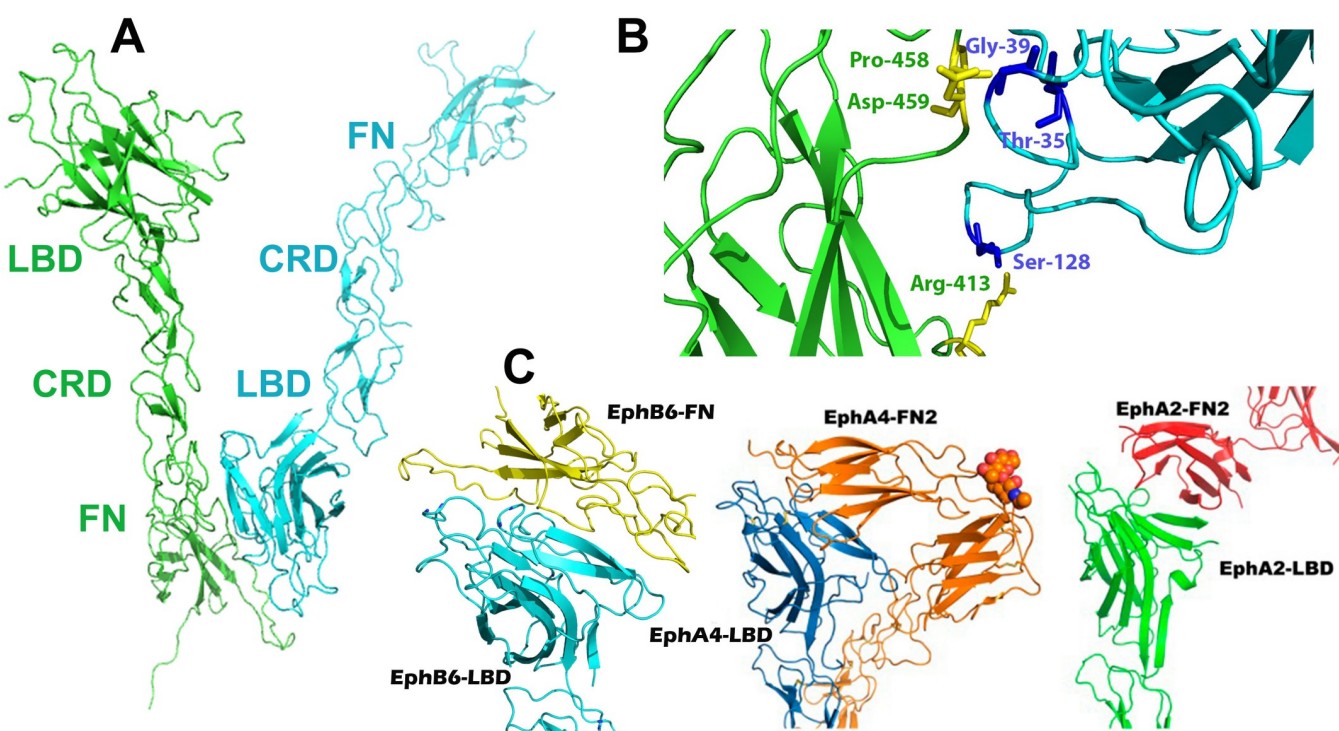

**Fig 6.** (A) Head-to-tail interactions in EphB6-ECD. Similar to the previously published A-class ectodomain structures, the LBD (cyan) and FN (green) domains of neighboring EphB6 molecules form an interface in the unliganded receptors. (B) The zoom-in shows the interacting amino acid residues. (C) Comparison of the head-to-tail interactions in EphB6-, A4- and A2-ECDs. (Left) EphB6-LBD (cyan) bound to EphB6-FN of a neighboring molecule (yellow) in the crystals of the unliganded EphB6-ECD; (Center) EphA4-LBD (blue) bound to the EphA4-FN of a neighboring molecule (orange) in the crystals of unliganded EphA4-ECD; (Right) EphA2-LBD (green) bound to the EphA2-FN of a neighboring molecule (red) in the crystals of unliganded EphA2-ECD.

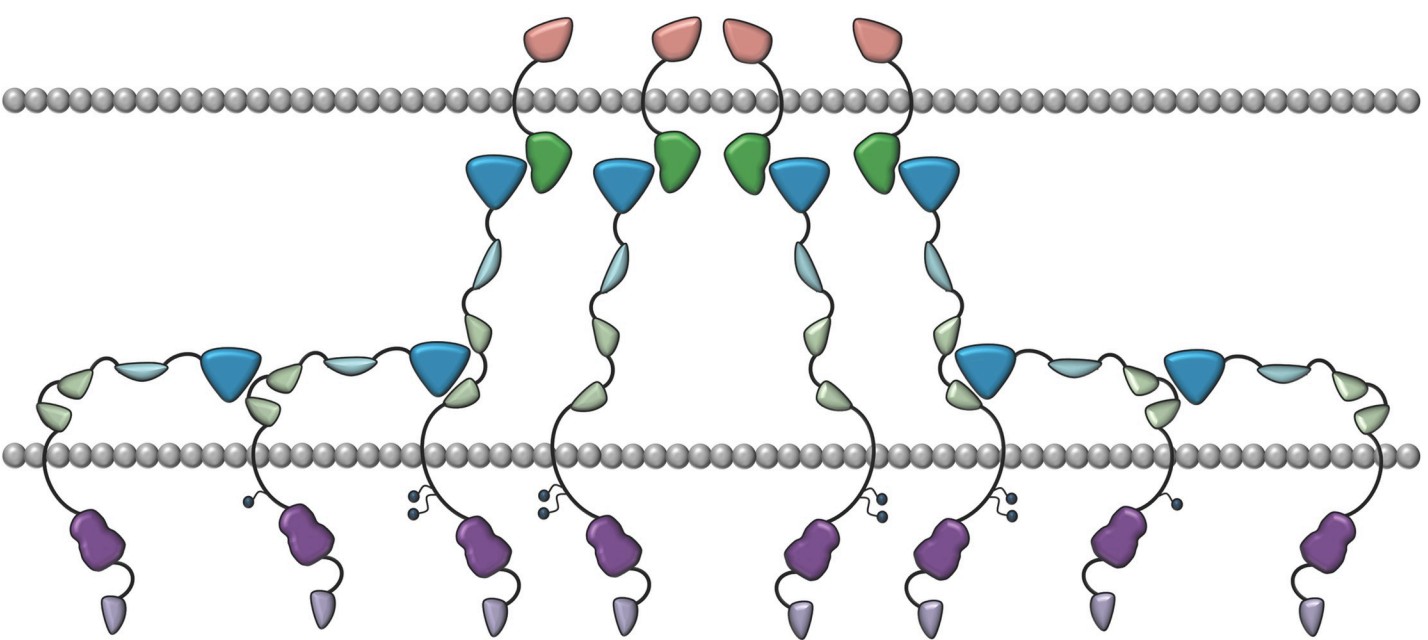

**Fig 7. Schematic representation of the head to tail Eph-ECD interactions.** The Eph receptors are in blue/green/purple and are interacting with the ephrins in green/orange. The LBDs of the ephrin-free Ephs (in blue) and the FN3 regions (in light green) of their neighbors are interacting. Intracellular, juxtamembrane Tyrosines are shown as small circles. They get fully phosphorylated only when biologically active heterotetramers and higher-order Eph/ephrin assemblies form after the initial ligand binding events. The cell membrane of the two interacting cells are in grey.

Glu-77, and Arg-106 on the LBD domain and Arg-454, Tyr-455, Asn-504, and Thr-507 on the FN domain [58]. The comparison of the structures shown in Fig 6C clearly highlights the difference in size between the LBD/FN interface of EphB6 and EphA4, the EphA4 interface being about three times larger than the EphB6 one. Importantly, it reveals the existence of a unique salt bridge between Glu-77 (LBD) and Arg-454 (FN) in EphA4, a major difference to EphB6 and, potentially, a crucial factor for the unique signaling characteristics of EphA4 [67]. As mentioned above, these 'head-to-tail' interactions may play a significant role in the fine-tuning of receptor clustering and cell to cell signaling, and previous mutagenesis studies of the influence of these interactions on the downstream (EphA4) signaling [58] have illustrated the regulatory effects that this interface mediates. Interestingly, the EphB6 receptor gets internalized and has only been shown to get downregulated after an exposure to a clustered ephrin-B2 ligand [68], while dimeric agonists have been found to cause the internalization and downregulation of EphA receptors [69, 70]. Future studies will show if differences in head-to-tail interface surface areas are affecting the dynamics of ligand-receptor complexes within the EphA and EphB subclasses.

## 4. Conclusions

To better understand signaling the molecular mechanisms of Eph signaling and to identify ectodomain interactions that can modulate this signaling, we expressed the ECD of the innately kinase-inactive EphB6 receptor, purified and crystallized it, and determined its high resolution structure. This represents the first reported structure of an Eph B-class ectodomain. Our data reveal that the overall architecture of EphB6 is similar to that of the A class receptors, but the individual EphB6 domains contain unique structural features and characteristics, which could help explain its exceptional signaling properties and biological function.

Similar to the EphA2 and EphA4 receptors, the unliganded EphB6-ECD is also involved in head-to-tail interactions between the LBD of EphB6 and the FN3 region of a neighboring receptor molecule (Fig 7). These interactions are likely to be involved in fine-tuning the biological output of EphB6 signaling both in normal and pathological conditions, and it remains to be determined if this mechanism is also used by the catalytically active members of the EphB receptor subgroup. The reported results are novel and relevant since EphB6 plays important roles both in normal physiology [19–24] and in human malignancies [25–45, 48]. Our work also provides insight into the molecular surface regions within the EphB6-ECD that may be used as targets for developing novel therapeutic reagents.

## Supporting information

**S1 Fig. Original 2D gel of EphB6-ECD.**
(TIF)

**S1 File. Crystal structure validation report.**
(PDF)

## Acknowledgments

The authors thank the Advanced Photon Source at Argonne National Laboratory for the use of their instrumentation to acquire crystallographic data.

## Author Contributions

**Conceptualization:** Emilia O. Mason, Dimitar B. Nikolov, Juha P. Himanen.

**Data curation:** Emilia O. Mason, Yehuda Goldgur, Dorothea Robev, Andrew Freywald, Juha P. Himanen.

**Formal analysis:** Emilia O. Mason, Yehuda Goldgur, Dorothea Robev, Andrew Freywald, Juha P. Himanen.

**Funding acquisition:** Dimitar B. Nikolov.

**Investigation:** Emilia O. Mason, Dorothea Robev, Andrew Freywald, Dimitar B. Nikolov, Juha P. Himanen.

**Methodology:** Emilia O. Mason, Dorothea Robev, Andrew Freywald, Dimitar B. Nikolov, Juha P. Himanen.

**Project administration:** Emilia O. Mason, Dimitar B. Nikolov, Juha P. Himanen.

**Resources:** Emilia O. Mason, Dimitar B. Nikolov.

**Software:** Emilia O. Mason, Yehuda Goldgur.

**Supervision:** Dimitar B. Nikolov.

**Validation:** Emilia O. Mason, Andrew Freywald, Juha P. Himanen.

**Visualization:** Emilia O. Mason, Yehuda Goldgur, Dimitar B. Nikolov, Juha P. Himanen.

**Writing – original draft:** Emilia O. Mason, Juha P. Himanen.

**Writing – review & editing:** Emilia O. Mason, Dimitar B. Nikolov, Juha P. Himanen.

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
