## [Decision Letter · Decision Letter 0]

30 Dec 2020

PONE-D-20-36209

Structure of the EphB6 receptor ectodomain

PLOS ONE

Dear Dr. Himanen,

Thank you for submitting your  manuscript to PLOS ONE. After careful consideration, based also on my own reading, we feel that it has merit but. However, few revisions are required for its final acceptance. Therefore, we invite you to submit a revised version of the manuscript that addresses the points raised during the review process.

We look forward to receiving your revised manuscript.

Kind regards,

Alessio Lodola, PhD

Academic Editor

PLOS ONE

Journal Requirements:

3.Thank you for stating the following in the Acknowledgments Section of your manuscript:

"This research was supported by funding from the National Institute of Health, The MSKCC Tri-Institutional Therapeutics Discovery Institute, The MSKCC Functional Genomics Initiative, and the Experimental Therapeutics Center of Memorial Sloan- Kettering, support from Mr. William H. and Mrs. Alice Goodwin and the Commonwealth Foundation for Cancer Research. X-ray diffraction studies at the Advanced Photon Source at Argonne National

Laboratory were supported by NIH (P41 GM103403) and by the U.S. DOE (Contract No. DE-AC02-06CH11357)."

 "D.B.N., R01-NS038486, National Institutes of Health, www.nih.gov; The funders had no role in study design, data collection and analysis, decision to publish, or preparation of the manuscript."

Reviewers' comments:

Reviewer's Responses to Questions

**Comments to the Author**

1. Is the manuscript technically sound, and do the data support the conclusions?

Reviewer #1: Yes

Reviewer #2: Yes

2. Has the statistical analysis been performed appropriately and rigorously? 

Reviewer #1: N/A

Reviewer #2: Yes

3. Have the authors made all data underlying the findings in their manuscript fully available?

Reviewer #1: Yes

Reviewer #2: Yes

4. Is the manuscript presented in an intelligible fashion and written in standard English?

Reviewer #1: Yes

Reviewer #2: Yes

5. Review Comments to the Author

Reviewer #1: The paper describe the structure of EphB6-Ectodomain which was determined through X-ray crystallography.

The findings are particularly interesting since EphB6 is the only kinase dead Eph receptor.

The paper is well conducted and well-written.

MINOR:

pag 11 line 251: A "super-serine" loop has been described but it is not reported in Figure 5

Reviewer #2: This is a very informative work aimed at providing a better understanding of the molecular mechanisms of Eph signaling, by solving the atomic resolution structure of the EphB6 receptor ectodomain, which is also the very first Eph B-class ectodomain crystal structure. While the overall architecture is very similar to those previously solved ectodomains of the A class receptors, some unique features here identified that could correlate with biological function, speculatively. Apart from some smaller differences, one common feature of ectodomains that has been reported is the possible interaction between the FN3 domain on one molecule and the LBD of another. This head-to-tail interaction has also been observed in this EphB6 crystal structure. The authors speculate that subtle differences in these interfaces could account for difference in biological activity, for example receptor activation. The work presented is novel (first even EphB ectodomain structure solved) and well described. Given that a few papers now claim that such interaction may be important in pre-clustering the unligated receptors, it would have been much stronger, in my opinion, if these intermolecular interactions between FN3 and LBD were isolated and more quantitatively characterized. Some EphA receptors get internalized by exposure with dimerized ligands (i.e. EphA2 with ephrinA1-Fc or potent dimeric ligands such as 135H12, i.e. Salem et al., Pharmaceuticals 2020, 13(5), 90), while this does not seem to happen to others (i.e. the EphA4 for example). Could it be, perhaps, that the ability of a given receptor to internalize after interacting with agonistic agents is linked to this FN3-LBD interactions? Does the EphaB6 get internalized after exposure to dimeric agonistic agents? At the least a comment in this regard would be useful and enrich the discussion/conclusions. Again, this is an excellent account.

6. PLOS authors have the option to publish the peer review history of their article (what does this mean?). If published, this will include your full peer review and any attached files.

Reviewer #1: No

Reviewer #2: No

---

## [Author Response · Author response to Decision Letter 0]

1 Feb 2021

Dear PlosOne Editors, 

Enclosed, please find our revised manuscript entitled “Structure of the EphB6 receptor ectodomain” that we are submitting for publication in PlosOne. We would like to thank the reviewers for their thoughtful comments and suggestions. Indeed, we believe these suggestions have made the manuscript stronger. We were happy to read that the reviewers find the work “well documented, well-written, and informative”. We made every effort to address the specific suggestions of the reviewers to modify Figure 5 and to comment on the internalization of EphB6 receptor upon the interaction with agonistic agents/ligands. We hope these changes are satisfactory to allow the publication of our manuscript. 

All revisions that we have made to the text are listed below: 

1) We have added an arrow showing the position of Ser-148 in Figure 5. This is the residue preceding the ‘Super-Serine’ loop. Since the loop is not structured in our model, this is, we feel, the best way to indicate that the loop, indeed, is part of the initial ligand-binding module, while the loop 125-130 is located on the opposite side of the EphB6-LBD, on the surface area that participates in the formation of the Eph/ephrin heterotetramers. 

2) We have modified the Figure 5 legend to indicate the change we made to this figure. 

3) We have added references and sentences at the end of the Results section to emphasize that both EphB6 and EphA2 get internalized upon ligand/agonist interaction, though possibly with different dynamics. 

4) We have also added a sentence on the possibility that the changes in the internalization dynamics might be the result of the different sizes of the head-to-tail interaction surface areas. While we are currently experimenting further on these homotypic interactions, a precise quantification of the interactions needs a new comprehensive study. 

Please let us know if there is any other information or materials you would need.

Sincerely yours, 

Juha-P. Himanen 

Sloan-Kettering Institute 

New York 

himanenj@mskcc.org

---

## [Editor Report · Decision Letter 1]

5 Feb 2021

Structure of the EphB6 receptor ectodomain

PONE-D-20-36209R1

Dear Dr. Himanen,

We’re pleased to inform you that your manuscript has been judged scientifically suitable for publication and will be formally accepted for publication once it meets all outstanding technical requirements.

Kind regards,

Alessio Lodola, PhD

Academic Editor

PLOS ONE
---

## [Editor Report · Acceptance letter]

18 Mar 2021

PONE-D-20-36209R1 

Structure of the EphB6 receptor ectodomain 

Dear Dr. Himanen:

I'm pleased to inform you that your manuscript has been deemed suitable for publication in PLOS ONE. Congratulations! Your manuscript is now with our production department. 

Kind regards, 

on behalf of

Dr Alessio Lodola 

Academic Editor

PLOS ONE